# Emotional Intelligence as a Predictor of Prosocial Behaviors in Spanish and Colombian Older Adults Based on Path Models

**DOI:** 10.3390/healthcare10020284

**Published:** 2022-02-01

**Authors:** Manuel Martí-Vilar, Carlos Andrés Trejos-Gil, Juan Diego Betancur-Arias

**Affiliations:** 1Department of Basic Psychology, Faculty of Psychology and Speech Therapy, University of Valencia, 46010 Valencia, Spain; 2Department of Social Communication, Faculty of Communication, Advertising and Design, University Catolic Luis Amigó, Medellin 05001, Colombia; carlos.trejosgi@amigo.edu.co; 3Faculty of Psychology and Social Sciences, University Catholic Luis Amigo, Medellin 05001, Colombia; juan.betancurri@amigo.edu.co

**Keywords:** emotional intelligence, prosocial behavior, path, confirmatory factor analysis, structural equation modeling

## Abstract

Emotional intelligence (EI), empathy, and prosocial behavior (PB) are widely studied in the early stages of life, up to adolescence. However, there have been few studies in older adults. Using a multivariate methodology, exploratory factor analysis (EFA) and confirmatory factor analysis (CFA) were applied with structural equation models (SEM) in 271 older adults in Colombia, along with a Spanish sample made up of 139 adults over 60 years of age, for a total sample of 411 older adults. The results confirmed that EI, as measured with the WLEIS, scale is the best predictor of PB, with excellent adjustment criteria (GFI = 0.99; CFI = 0.98; RMSEA = 0.064; PCLOSE = 0.28; SRMR = 0.023; AIC = 57.30). It is concluded that the path model yielded a reliable predictive explanation of PB, including EI as a key variable that explains prosocial tendencies.

## 1. Introduction

Predictors of prosocial behavior (PB) have been investigated in many populations over the last decade or so [1,2,3,4]. This is unsurprising given the importance of knowing which factors are associated with the antithesis of antisocial behavior, because the accumulation of evidence on factors involved in antisocial behavior is greater than that from studies of the factors that drive PB.

One of the variables with the greatest structural weight in predicting PB in any population group is emotional intelligence (EI) [5,6,7,8,9,10,11]. Studies of EI support the role of cognitive, affective–emotional, and motivational modulators in PB that, when linked to balanced emotional programs, can allow forms of interpersonal social interaction that enhance well-being and life satisfaction.

The concept of EI has its roots in the scientific work on social intelligence carried out by Thorndike in 1920 [12]. This author defined social intelligence as the ability to understand and manage interactions with other people and to act wisely in human relationships. Much later, Salovey and Mayer (1990) [13] defined EI as “the ability to manage our emotions and those of others, discriminate between them and use this knowledge or emotional information to guide our own actions and thoughts”. They proposed the use of this name and advocated scientific work focused on explaining and predicting the processing of emotional information. They established a model called the “ability model”, that integrates a triad of skills: emotional perception and appreciation, emotional regulation, and the use of EI. This initial conception included the cognitive and affective spheres of emotion, but also treated emotional intelligence separately as a higher cognitive skill. It is also worth noting that, at that time, the influence of motivational factors on EI was under-researched.

After this, mixed models of EI appeared, which included skills and competencies, but also considered personality traits and affective dispositions, among other variables. Two of the most representative authors of this perspective are Bar-On [14,15] and Goleman [16], who proposed a “mixed model”. However, when considering these authors’ studies, it is pertinent to take into account their participants’ life history and behavioral record in terms of aggression, since an analysis of EI cannot be made in isolation before issuing a mental health diagnosis, without considering such disruptive episodes [17].

EI is assumed to serve as the mediating dimension between skills, attitudes, aptitudes, and socio-affective competencies, in relation to stable personality traits or patterns. EI contributes to social adaptation, and cognitive aspects of EI play a fundamental role in the establishment and maintenance of such thoughts, motivations, and behaviors as empathy, prosocial moral reasoning, social values, and the self-regulation of violent behaviors [18,19]. All these aspects are related to an individual’s levels of PB.

The regulation of emotions works differently in older adults versus younger adults, since the latter’s goals are oriented towards the future, in terms of acquiring knowledge and having a desire to learn, while in older adults, goals tend to be oriented to the present and focused on improving their feelings and emotions [20].

### 1.1. Emotion and Prosocial Behavior

Emotions are processes that are experienced in the body/brain, and the aspects of thought linked to these are called feelings. Emotions are instinctual repertoires that favor positive dynamism in interpersonal relationships. Emotions are “reasonably complex action programs […], triggered by an identifiable object or an event, an emotionally competent stimulus” [21] (p. 131), which tend to promote well-being, within-group cooperation, and, ultimately, survival. According to Damasio, emotions exist in groups, and he defends the idea that an emotion appears before one is conscious of it. In addition, he distinguishes between universal primary emotions on the one hand, and social emotions that can be modified by the specific environments and experiences of individuals, on the other. Damasio states that these are the same emotions: what varies is the mechanism that triggers them. This is known as the somatic marker hypothesis of emotion, which attempts to explain both the physiological mechanisms that produce emotions and their ultimate functions, as determined by our evolutionary biological inheritance [21].

Positive empathic emotions arise in the framework of social interactions and require the ability to empathize with others, implying a sensitivity to the emotions and needs of other people. They are therefore a powerful motivator of prosocial behavior [22,23]. Empathy is linked to positive emotionality, and both influence the development of prosociality from childhood onwards. Empathy is presented as the main motivating factor of prosocial behavior, especially in its emotional dimension (empathic concern), but also in its cognitive dimension [24]. There is already a lot of research that can relate this construct, together with moral judgement, regulation, and emotional stability, with prosocial behavior [25,26,27].

In the same way, emotional intelligence is also considered a fundamental component of prosocial behavior. However, the literature is still scarce, and scientific studies that relate EI with empathy in older adults refer to the need to delve into purposes like this [11,28]. As a result, the relationship between empathy and prosociality will not be the main focus of this article; instead, the link between emotional intelligence and prosocial behavior will.

### 1.2. Emotional Intelligence and Prosociality

Prosociality refers to a set of behaviors characterized by intentions to help others without receiving anything in return [29]. Like empathy, prosociality allows us to participate in the emotions and feelings of other people, to the point of including them in our cognitive, emotional, and motivational spheres of behavior. Prosociality has been studied using different theoretical explanatory models, and it has been concluded that it is acquired throughout different stages of life, especially in early ages, such as adolescence and young adulthood, and that it is related to emotional and cognitive development [22,23].

However, prosocial behavior in adulthood represents one scientific challenge in social and cultural research, due to the scarcity of research. According to previous studies [30,31,32], prosociality may be influenced by different variables that predict a positive evolution trough the transition from youth to adulthood, such as the individual aspects of biological potential, education, the development of social values, civic competences, social interest groups, economic activities, family topology, individual psychological aspects, and characteristics of personality, etc. According to a search in Web of Science, which was carried out in November 2021, using the terms “prosociality” AND “older adults”, the literature among prosocial behavior in this age group is exponentially increasing. Apparently, older adults show more prosocial behavior than young adults [33,34,35], even during the pandemic [36]. It seems that the prosocial behavior in older adults refers to a stabilization of social conditions suitable to serve the other from unconditionality, moderation, the desire to contribute, sensitivity towards the evil of others, and a prevailing need to generate personal and social well-being in the different contexts of the life of the individual. However, further research on this topic is needed.

Among the personal variables that can influence prosocial behaviors, emotional intelligence (EI) has received much attention in the study of attitudinal and social competencies [37,38,39]. A link between EI and prosociality is justified, since this type of intelligence, as defined by Mayer and Salovey [40], refers to the ability to perceive, assess, and express emotions in a way that is sensitive to the contextual realities of situations and people, the emotional facilitation of thought, and the comprehension and reflexive regulation of emotions. Thus, the unification of emotion and reason in EI allows a more effective, in-depth reflection on emotional aspects of thought [40]. EI influences the social integration and effective adaptation of people to their social environment through prosocial behaviors [28,41].

### 1.3. The Current Study

The main objective of the current study is to analyze the predictive variables of prosocial behavior in older adults. To do so, [18] was taken as a reference. However, in this study, the sample characteristics were modified such that the age group changed from young to older adults, and a population from Colombia was included. Additionally, the analyses followed a path statistical model. Along these lines, the following research hypotheses were proposed.

Cultural variables play an important role as predictors of prosocial behavior [42,43]; however, the literature is not as conclusive as it seems. For instance, there is also evidence that concludes individualism is better than collectivism when predicting prosocial behaviors [44], and constructs, such as uncertainty avoidance and future orientation, are negatively correlated with prosocial behaviors [10]. Emotional intelligence, on the other hand, showed few, but more cohesive, results, as they were explained in the introduction. However, emotional intelligence is expected to explain prosocial behaviors much more robustly, as there is consistency across numerous articles about the important role of emotional intelligence as a predictor of prosocial behavior [45,46,47,48].

**Hypothesis** **1** **(H1).**
*Emotional intelligence predicts prosocial behavior more significantly than cultural variables in older adults do.*


Two scales of emotional intelligence were used in order to achieve a better understanding of the relationship between this construct and prosocial behavior. Each scale was used as a representation of each of the two most relevant theories of emotional intelligence—the ability model [13] and the mixed model [14,15]. The ability model was represented by “Wong-Law’s Emotional Intelligence Scale” (WLEIS) [49] and the mixed model by the EQi-C Emotional Intelligence Scale [14]. It was predicted that the WLEIS would be more related with prosocial behavior, due to its nature of describing traits that are assumed to be stable among people, while the EQi-C scale is more dependent on the individuals’ specific background.

**Hypothesis** **2** **(H2).**
*The WLEIS scale shows greater precision than the EQi-C scale in predicting PB in older adults.*


Finally, masculinity and femininity dimensions were compared as they seemed to play different roles in the expression of prosocial behavior.

There is research that shows that there is a particularly strong gender pressure with respect to prosocial values in early adolescence [50]; however, as this variable is observed at older ages, the trend in gender differences goes down. The study by Nielson et al. [51 found that in older samples there were no differences in levels of defensiveness and physical helping behavior toward friends, indicating that there may be fewer gender differences as people age. On the other hand, there are theories that argue that culture has always played a key role in the greater prevalence of prosocial behavior in men. Thus, they argue that, from the beginning of recorded history, men, in general, performed the hardest and most strenuous jobs for the benefit of others and themselves [51]. These differences in prosociality have been able to be maintained throughout history because of cultural pressure that, until the last century, forced men to work to support their families [51].

**Hypothesis** **3** **(H3).**
*Masculinity shows positive effects on prosocial behavior.*


## 2. Material and Methods

### 2.1. Participants

The total sample consisted of 434 participants, recruited via convenience sampling in gerontology centers, and 33 were excluded due to missing or invalid data, or errors in filling in the survey. Therefore, the final sample consisted of 411 participants, 260 from Colombia (68.4% women) and 151 from Spain (54% women). In total, the sample was composed of 63.5% women and 36.5% men. With regard to age, 47.9% were between 60 and 65 years; 36.3% were between 66 and 70 years; 11.9% were between 71 and 75 years; 2.9% were between 76 and 80 years; 1% were between 81 and 88 years of age.

### 2.2. Material

Four instruments were implemented in this study. Two of them measured emotional intelligence, one measured the individual cultural factors, and the last one was applied to measure the prosocial behavior. The final aim of this procedure was to measure the degrees of relationship, prediction, and incidence of the two emotional intelligence scales among the rest of variables, such as cultural factors and prosocial behavior.

The first scale which measured emotional intelligence was “Wong-Law’s Emotional Intelligence Scale” [49], in a Spanish adaptation by Fernández-Berrocal et al. (2004) [52]. This instrument is based on the model of ability [13]. This scale contains 16 items grouped into 4 factors: (1) emotional self-assessment (the ability to understand and express personal emotions); (2) emotional evaluation of others (the ability to recognize emotions in other people); (3) use of emotions (the ability to redirect emotions into productive behaviors); (4) emotional regulation (the ability to control one’s emotional responses). Its responses were presented as a Likert-type scale of 5 options that ranged from “totally disagree” to “totally agree”.

The second instrument used to measure emotional intelligence was taken from the mixed model of Bar-On [14]. The EQi-C Emotional Intelligence Scale [14] was used, in an abbreviated Spanish adaptation by López-Zafra et al. (2014) [53]. This includes 28 items grouped into 4 factors: (1) impersonal intelligence (the understanding of other people’s emotions); (2) adaptability (problem solving and change management); (3) stress management; (4) intrapersonal intelligence (expression and understanding of one’s own emotions). The answers were collected through a Likert-type scale of 5 options from “never” to “always”.

Individual cultural factors were measured using “The scale of cultural dimensions at an individual level” [54], in a Spanish adaptation by Hernán-Rodríguez (2011) [55]. This consists of 23 items grouped into 5 cultural dimensions: (1) collectivism; (2) avoidance of uncertainty; (3) power distance; (4) masculinity; (5) orientation towards the future. The response format was Likert-type scale of 5 options that went from “totally disagree” to “totally agree”. Items (1,3,8,9,10,12,17) were eliminated from the original scale due to their low loads with extractions below 0.3 according to the theory; being reduced in its components by eliminating items, it was made up of the 16 remaining items, obtaining only 3 factors from the original 5 (Table 1—culture).

Finally, prosocial behavior was measured with the PBS prosociality scale [56], an instrument of prosocial behavior in adults that measures assistance, reliability, and sympathy. The total score makes it possible to ordinally compare participants according to their overall level of prosociality. The responded were provided of a Likert-type scale of 5 options, that ranged from “never” to “always”.

A demographic characterization section with four subsections (sex, age, marital status, and country of residence) was also implemented for a total of 87 variables, split between the 4 scales and the 4 demographic sections.

### 2.3. Procedure

For the selection of participants in Colombia, the study had the support of the management of local programs aimed at older adults who were physically and mentally healthy and whose neurocognitive profile corroborated this. In the case of Spain, the questionnaire was applied to a convenience sample. The sample was screened following the same inclusion criteria as Colombia, so the participants’ families were asked to make sure the older adults were both physically and mentally healthy.

In both settings, participants were informed of the nature and purpose of the research, and their voluntary participation was requested. Informed consent was given during an interview that allowed both the researcher and the elderly person to sign the consent form and resolve any doubts. The questionnaires were filled out following a paper-and-pencil procedure in both countries. Volunteers were trained on how to instruct the participants to fill in the questionnaires, so they followed the same procedure. Additionally, the assistants were asked to stay with the participants to answer any question that may arise. This procedure in both countries was carried out in the second half of 2019.

### 2.4. Data Analysis

The instrument was subjected to various preliminary statistical analyses. The internal consistency [57,58] was analyzed using Cronbach’s alpha statistic (α = 0.878) to estimate dependence, independence, and reliability indicators for the 83 items. The adequacy of the correlation matrix was verified in order to see its possible factorization under the multivariate technique of the Kaiser–Mayer–Olkin (KMO) criterion, along with Bartlett’s sphericity test, the maximum likelihood extraction method, and promax rotation analysis. A combination of exploratory factor analysis (EFA) and confirmatory factor analysis (CFA) was deployed. In the case of the EFA, following Hair et al. (1999) [59] and Pérez et al. (2013) [60], we proceeded to eliminate those variables with low extractions, with the aim of improving the KMO, and then applied EFA once again to the remaining variables. This made it possible to eliminate variables 9 and 14 of the prosociality scale; 1, 3, 8, 9, 10, 12, and 17 of the culture scale; and 5, 7, 8, 9,12, 13, 14, 15, 16, 20, 24, and 27 of the EQi-C scale, on account of their coefficients being less than 0.400 in each case. The entire instrument with the 62 items of the 4 scales together obtained a good reliability (α = 0.878).

IBM’s Statistical Product and Service Solutions software (SPSS Statistics^®^ version 25) was used for statistical significance analysis, together with the AMOS^®^ v26 plug-in [61], for the purpose of proceeding to CFA with the maximum likelihood method and robust statistics on the models proposed in this study. This was in accordance with the adjustment recommendations suggested by Gaskin and Lim (2016) [62], DiStefano and Hess (2005) [62], and Hu and Bentler (2009) [63]. The reference adjustment indices were based on those proposed by these same authors, namely: normed chi-square χ^2^/df < 5; non-normed fit (NNFI) > 0.90; root mean square error of approximation (RMSEA) < 0.08 and <0.06; goodness-of-fit index (GFI) and comparative fit index (CFI) > 0.90; standardized root mean square residual (SRMR) < 0.08; *p*-value for perfect fit test (PClose) > 0.05.

## 3. Results

Items in the initial results with extraction coefficients < 0.4 were detected and eliminated [64,65,66,67]. Accordingly, in order to improve the extraction of variance, 2 items were eliminated from the prosociality scale, 7 items from the culture scale, and 12 from the EQi-C scale; the resulting extractions expressed the proportion of the variance in the variables explained by the extracted factors. The results in the KMO suitability index, in all cases, were greater than 0.6 and close to 1, while the Bartlett test of sphericity was approximately 0.000 (*p* < 0.05; see Table 1 and Table 2). Therefore, the application of the exploratory factorial model with all its extracted variables was adequate for explaining the phenomenon, and the analysis indicated that it was feasible to apply EFA to explain associations in the data.

When applying the multivariate maximum likelihood technique with promax extraction in the EFA of the 4 scales, 3-factor solutions were produced for the prosociality scale (PBS), the culture scale, and EQi-C, and a 4-factor solution in the case of the WLEIS scale, with extracted variance figures of 42.5%, 39.4%, 37.5%, and 52.7%, respectively. The factors that best explained each scale were called by the researchers “Active predisposition towards help (AP)” in the prosociality scale (α = 0.877), the factor 1 explaining 34.6% of the variance in that scale; “Planning without assuming risks in the future (PaRF)” in the culture scale (α = 0.750), the factor 1 accounting for 22.2% of the variance in that scale (Table 1); “Self-control/Stress (ScS)” in the EQi-C scale (α = 0.709), the factor 1 explaining 18.7% of the variance in that scale; and “Emotional regulation” (Table 2), in the factor 1 accounting for 37.3% of variance in the WLEIS scale (α = 0.899). Internal consistency reliability was adequate for all multivariate factors.

To validate the predictive variables of prosociality according to the described scales, the analysis of 4 confirmatory models was carried out using CFA. For each scale, a structural equation model (SEM) was constructed, formed as follows. In the case of the prosociality scale, it was composed of the following latent variables: AP with 8 observed variables, empathetic conduct (EC) with 3 observed variables, and personal motivation (PM) with 3 observed variables (Figure 1). For the culture scale it included the latent variables PaRF with 6 observed variables, masculinity power (M-P) with 7 observed variables, and discipline (D) with 3. The SEM for the EQi-C scale included the latent variables AE with 7 observed variables, interpersonal (I) with 5 observed variables, and coping/adaptation/adaptability (CAA) with 4 observed variables. Finally, that for the WLEIS scale included all 4 of its latent variables emotional regulation (ER), intrapersonal emotional intelligence (iEI), interpersonal emotional intelligence (IEI), and self-motivation (A), with 4 observed variables each.

Within the 4 models in Figure 1, high covariances were evidenced between all latent variables of prosociality, highlighting the best covariance between AP and EC and in the second instance between AP and PM. In the same way, among the latent variables of WLEIS, the best indices between iEI and A stood out, in the second instance between the latent variables ER and A and with the same iEI and IEI value with coefficients of 0.69, 0.67, and 0.67, respectively. In a covariance between the latent variables of both culture and EQi-C showed better coefficients in PaRF and D for culture, and I and CAA with coefficients 0.66 and 0.73, respectively (Figure 1). The aforementioned constructs and those that present positive coefficients of covariance indicate optimal adjustments [58,63].

Regarding the adjustments of the presented models, for the value of normed χ^2^ (χ^2^/df) in Table 3, it can be observed that in no case is it less than 1 (which would indicate over-adjustment) or requiring re-adjustment, taking into account that all values are between 1 and 3. The values of GFI and CFI are all greater than 0.90, exceeding the minimum decision value for a good adjustment (0.90). The SRMR values are excellent (<0.06) in the EQi-C, WLEIS, and prosociality models, and acceptable (>0.06 and <0.08) in the culture model. RMSEA values are excellent (<0.06) in EQi-C and prosociality, and acceptable (>0.06 and <0.08) in culture and WLEIS. The PClose values are excellent (>0.05) only in prosociality, and acceptable (<0.05) in culture, WLEIS, and EQi-C. With the AIC statistic, the prosociality model, 192.62, indicates greater parsimony of the data than the other scale models.

Figure 2 shows that EQi-C directly affected prosociality to a lesser extent than the WLEIS scale did, with the EQi-C coefficient being 0.27 and the WLEIS coefficient being 0.34. The magnitude of the indirect effect of the EQi-C scale on prosociality is −0.005; in turn, the WLEIS scale also has a negative indirect effect on prosociality, with a magnitude of −0.01. The scale that most affects prosociality, both directly and indirectly, is the WLEIS scale. The culture scale affects prosociality in the opposite way, due to its direct effect having a negative magnitude (−0.04). Table 4 shows the results of the MANOVA carried out to study differences in prosociality with respect independent variables (culture, EQi-C, WLEIS). In all analyses there were significant differences (*p* ≤ 0.001) in the total score of prosociality (*F* = 37.083, *p* < 0.001).

In view of the results obtained through the proposed path model, the direct and indirect effects of EQi-C and WLEIS on prosociality indicate that the WLEIS scale is the one that has the greatest influence as a predictor of prosociality scores. The control variable with the best explanatory contribution is the age variable, with a direct effect of 0.10, although its effect on prosociality is not very important due to its low contribution. The covariance between the control variables and the exogenous variables is presented mostly negatively, and ultimately does not affect the model since they are control variables, but it does reflect a slight increase in the total variance of the exogenous variables over the endogenous variables.

With regard to the metric invariance, there are no significant differences between the groups of Colombia and Spain to prosociality; with a confidence of 90% and a margin of error of 10%; chi-square 23,187, degrees of freedom (Df = 14); *p*-value (*p* = 0.057). There are significant differences between the groups of Colombia and Spain in relation to culture; chi-square 14.8, (Df = 16) (*p* = 0.539). There are significant differences between the Colombian and Spanish groups in relation to the IEWLEIS; chi-square 14.8, (Df = 22.1) (Df = 16) (*p* = 0.14). There are significant differences between the groups in Colombia and Spain in relation to the IEEQ; chi-square 7, (Df = 16) (*p* = 0.973).

Globally, there is partial scalar invariance for the culture scales; chi-square 9.158, (Df = 8) (*p* = 0.329); prosocial behavior; chi-square 11.246, (Df = 7) (*p* = 0.128); WLEIS chi-square 13.444, (Df = 7) (*p* = 0.062), and EQi-C chi-square 11.028, (Df = 11) (*p* = 0.441).

The proportion of the extracted variance explained by the model is 26%, representing moderate significance. This suggests that other variables should be incorporated to increase the variance explained by the model. The fit indices of the path model are excellent and acceptable according to each criterion used to test the proposed theory (Table 3). The normed *χ^2^* value (*χ^2^*/df) is 2.651, which is less than 3; the criteria of GFI (0.99) and CFI (0.987) are both close to 1 for an excellent fit; SMRS is 0.023, which is identified as acceptable, as is the RMSEA (0.064) criterion. Similarly, the criterion of PClose (0.280) is excellent since, according to the criterion, it must exceed 0.05 when testing the null hypothesis; and finally, the Acaike information criterion (AIC = 57.303) indicates parsimony of the data.

The most relevant direct effect [56] on CP was the effect exerted by the WLEIS scale. (WLEIS → PB = 0.34). The EQi-C, country, and culture scales also directly influenced PB with path coefficients of 0.27, 0.04, and −0.04, respectively, with country and culture being non-relevant influences. The model explained 26% of the variability of the dependent variable. However, the relevance of the scales that were analyzed is evidenced in the moderate path coefficient of the magnitude of indirect effects. The WLEIS scale was substantial (WLEIS → PB = 0.33) and a relatively low magnitude of indirect effect was relatively low on the scale (EQi-C → PB = 0.2648).

## 4. Discussion

The objective of this study was to explain the predictors of PB in older adults. Following a factor analysis to regroup the items of PB, WLEIS, EQi-C, and cultural dimensions, and a path model, it was confirmed that emotional intelligence measured with the WLEIS was the best predictor of PB. The results of this study coincide with those obtained in another developmental stage, early adulthood [18]. The levels of reliability and validity of the instruments used to measure PB can be considered high due to the rigorous EFA, CFA, SEM, and path analysis. This study is considered an innovation in this field because previous research focused on the study of PB does not use such rigorous forms of analysis, nor have they studied a sample of older adults [39,68].

Several reviews [41,69] showed that cultural differences are a determining principle that differentiates people in their behavior. In this specific case, and in view of the results obtained, this variable does not contribute as much to prosocial behavior as emotional intelligence does. Thus, the first hypothesis was confirmed. This conclusion is not surprising due to the fact that there is not a consensus of the interaction of their factors along the literature [10,44], so a direct effect among cultural variables and prosocial behavior cannot be found. However, this effect seems to be clearer when prosocial behavior is mediated by emotional intelligence, at least in adolescents [18,46]. That is why it can be concluded that emotional intelligence and its concomitant variables are better predictors of prosocial behavior than cultural variables.

Regardless of the socio-cultural programs that an individual uses, the WLEIS is an instrument that is a major predictor of prosocial behavior in different parts of the world (excluding culture as a contextual philosophy) and also in different age groups. This may be because the WLEIS is based on an ability model. This model is more influenced by situational factors than other models, based on dispositional personality factors, which are designed to predict typical behavior [70]. There are several studies that have revealed the predictive role of certain situational variables in prosocial behavior. Among these variables are the following factors: ambiguity of need, severity of need, physical appearance of the victim, weather conditions, similarity to the victim, friendship or involvement, number of bystanders, location (urban or rural), and cost of helping—these are all situational variables [71,72,73]. In addition, there is a classic study that demonstrated that the situational variable bystander effect exerts a particular influence. This research showed that the more people witness and observe an emergency situation, the less likely one of them is to perform a helping behavior [74] (Darley and Latané, 1968).

Masculinity in culture shows positive effects on prosocial behavior (*p*-value = 0.000). This factor, when evaluated through cultural patterns, also favors the development of prosocial behaviors [75], where masculinity is related to little emotional expression. However, this disparity may be due to differences in the conceptualization of roles [76], since high scores of masculinity in women (androgyny) are associated with higher prosocial behavior [77]. These results confirm the third hypothesis and are in line of the work of Nielson et al. (2017) [51] and its statement, which asserts that the differences between masculinity and femininity on their interaction with prosocial behavior are reduced among older adults.

### 4.1. Limitations and Future Research

This study has some limitations. First of all, to find an equal number of participants would have been optimal for both countries, but the questionnaires were administered to 434 subjects and only 401 participants were validated. However, being a small difference, the analysis was not excessively affected. Secondly, this study is cross-sectional. In contrast to longitudinal, cross-sectional studies have a lower statistical power [78]. Thirdly, the selection of participants was incidental, which means that they came from different backgrounds and no previous screening was made. Only the conditions of being both mentally and physically healthy were assessed. However, in the Spanish sample, this was only verified by their relatives. It would be interesting if they also had an updated diagnosis which made sure these conditions were achieved. Furthermore, this study can be classified as pure quantitative. Mixed model studies (both quantitative and qualitative) have been recently highly recommended as they are considered to have a higher inference quality [79,80,81], as well as giving the chance for the readers to produce meta-inferences [82]. Lastly, self-report biases could be affecting the results. For instance, social desirability, insufficient effort, or response patterns, etc.

For future research, a comparison between empathy and emotional intelligence as predictors of PB is proposed, in order to analyze which of the two predicts prosociality to a greater extent in different samples. Moreover, it would be interesting to propose more studies that measure these variables among older adults, including longitudinal and mixed model approaches. Finally, it would also be advisable to work with participants from other countries to be able to identify if wider differences in culture than those found between Colombia and Spain can affect prosocial behavior, or if emotional intelligence provides predictive value that is truly independent of culture.

### 4.2. Practical Implications

This study can be considered as highly interesting concerning the novelty of the topic. Older adults have been studied from a negative or passive approach, emphasizing their role as a kind of victim [83], which is clearly not the objective of this study. The assessment of emotional intelligence and prosocial behavior among this sample locates this population in an active role, which is a good first step to work with. Intervention programs directed to increase their emotional intelligence through prosociality can be proposed, as long as these results showed a good connection among these variables.

Moreover, prosocial behavior and helping activities have been found to have a positive correlation with well-being among older adults [84]. For that reason, it can be considered quite a relevant topic to focus on. Furthermore, this assessment served to continue recommending the WLEIS when working with this age group. To sum up, this study sheds light on both the academic and the intervention fields, which need to be working hand in hand to provide real solutions and feasible improvements to the general population in order to enhance their living standards.

## 5. Conclusions

As a general conclusion, the results of this study show that, consistent with the finding of the empirical studies performed by Caprara et al. (2012) [22] and Ferguson et al. (2018) [85], prosocial behavior derive from stable personality patterns. The present study confirms the existence of emotional skills that are configured as distinctive features of older adults who have a marked tendency to behave in a prosocial manner, regardless of the cultural aspects that could shape this behavior.

## Figures and Tables

**Figure 1 healthcare-10-00284-f001:**
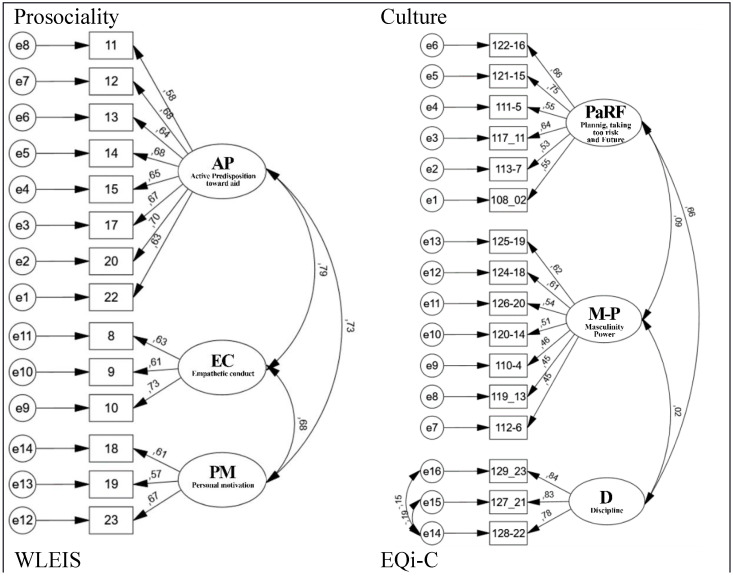
Structural models of the 4 scales: prosociality, culture, WLEIS, and EQi-C.

**Figure 2 healthcare-10-00284-f002:**
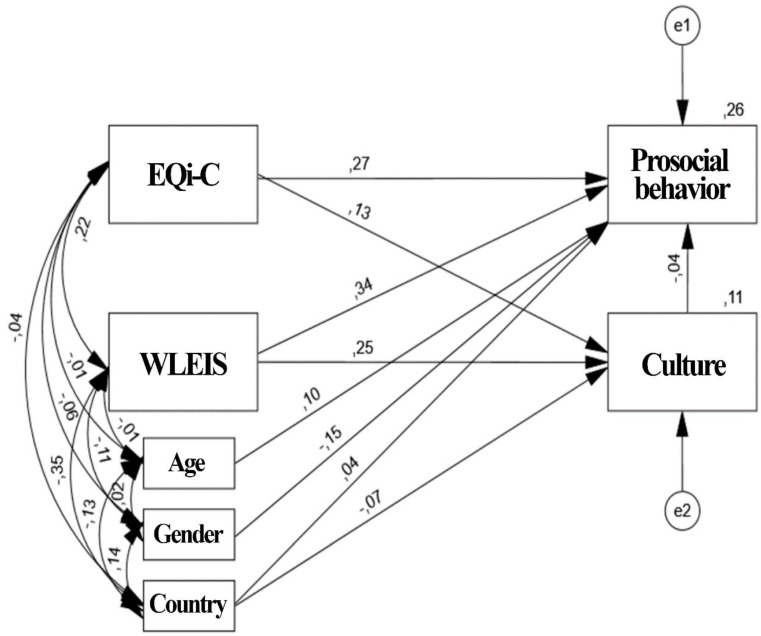
Path model of the hypothesized relationships between the dependent variable prosociality and EQi-C, WLEIS, and culture, with control variables, such as age, sex, and country.

**Table 1 healthcare-10-00284-t001:** EFA matrix–promax rotation–total variance explained. N = 411. Prosociality and culture.

Prosociality					Culture				
Item	AP	EB	PM	α	Item	PaRF	M-*p*	D	α
7	0.755			0.866	16	0.738			0.731
5	0.747			0.865	15	0.627			0.725
6	0.674			0.867	5	0.597			0.737
15	0.596			0.868	11	0.586			0.738
8	0.577			0.866	7	0.582			0.737
10	0.570			0.866	2	0.474			0.737
4	0.570			0.87	19		0.646		0.746
13	0.525			0.863	18		0.606		0.739
12		0.801		0.878	20		0.556		0.746
11		0.527		0.873	14		0.513		0.731
16		0.400		0.869	4		0.46		0.751
1			0.827	0.872	13		0.438		0.751
2			0.461	0.87	6		0.435		0.734
3			0.422	0.867	23			0.84	0.736
					21			0.838	0.735
					22			0.67	0.734
% Variance	34.66	4.21	3.64	42.51	% Variance	22.18	12.24	5.04	39.46
Auto value	2.47	0.300	0.26		Auto value	1.39	0.765	0.315	
Kaiser–Meyer–Olkin adequate sampling				0.918	Kaiser–Meyer–Olkin adequate sampling				0.818
Bartlett’s sphericity test			χ^2^	1761.23	Bartlett’s sphericity test			χ^2^	1819.73
			df	91				df	120
			Sig.	0.000				Sig.	0.000

AP—active predisposition to help; CE—empathic behavior; PM—prosocial motivation. PaRF—planning, not assuming risks and future; M-P—masculinity-power; D—discipline.

**Table 2 healthcare-10-00284-t002:** EFA matrix–promax rotation–total variance explained. N = 411 WLEIS and EQi-C.

WLEIS						EQi-C				
Item	ER	iEI	IEI	A	α	Item	AE	I	CAA	α
14	0.817				0.891	28	0.701			0.688
13	0.795				0.891	23	0.68			0.679
15	0.727				0.893	19	0.658			0.683
16	0.692				0.889	25	0.632			0.699
2		0.826			0.893	26	0.585			0.689
1		0.710			0.892	2	0.543			0.695
3		0.645			0.893	10	0.525			0.703
4		0.483			0.894	21		0.716		0.692
8			0.673		0.894	17		0.644		0.697
5			0.609		0.895	18		0.457		0.7
6			0.607		0.893	4		0.443		0.7
7			0.579		0.898	22		0.429		0.703
12				0.841	0.892	6			0.701	0.701
11				0.648	0.892	1			0.531	0.705
10				0.638	0.894	3			0.464	0.697
9				0.441	0.894	11			0.419	0.693
% Variance	37.19	6.97	4.40	4.07	52.65	% Variance	18.68	15.25	3.56	37.49
Auto value	2.32	0.43	0.27	0.25		Auto value	1.16	0.95	0.222	
Kaiser–Meyer–Olkin adequate sampling					0.89	Kaiser–Meyer–Olkin adequate sampling				0.825
Bartlett’s sphericity test				χ^2^	2791.24	Bartlett’s sphericity test			χ^2^	1575.63
				df	120				df	120
				Sig.	0				Sig.	0.000

ER—emotional regulation; iEI—intrapersonal emotional intelligence; IEI–interpersonal emotional intelligence; A—self-motivation; SS—self-control/stress; I—interpersonal; CAA—coping/adaptation/adaptability.

**Table 3 healthcare-10-00284-t003:** Structural model indices based on the CFA of prosociality, culture, WLEIS, and EQi-C.

Measurement	χ^2^	df	χ^2^/df	GFI	CFI	NNFI	NFI	RMSEA	LO 90	HI 90	PCLOSE	AIC	SRMR
EQi-C	213.49	101	2.11	0.94	0.92	0.91	0.86	0.052	0.042	0.062	0.34	283.49	0.0502
Culture	251.77	99	2.54	0.93	0.91	0.89	0.86	0.061	0.052	0.071	0.024	325.77	0.0625
WLEIS	251.64	98	2.56	0.93	0.94	0.93	0.91	0.062	0.052	0.071	0.02	327.64	0.041
Prosoc.	130.62	74	1.76	0.95	0.96	0.96	0.93	0.043	0.031	0.055	0.81	192.62	0.0356
Path model	5.30	2	2.65	0.99	0.98	0.86	0.98	0.064	0	0.132	0.28	57.30	0.023

χ^2^—chi square; df—degrees of freedom; χ^2^/df—normed chi square; GFI—goodness-of-fit index; CFI—comparative goodness index; NNFI—non-standard fit index; NFI—normed fit index; RMSEA—root mean square error of approximation; PClose—*p*-value for perfect fit test; AIC—Akaike information criterion; SRMR—standard square root residual index.

**Table 4 healthcare-10-00284-t004:** MANOVA results for the dependent variable prosociality, analyzing the effects of the independent variables culture, WLEIS, and EQi-C.

D.V.: Prosociality	Sum of Squares	Df	Mean Square	*F*	Sig.
Regression	6678.31	3	2226.102	37.083	0.000 *
Residue	24,432.1	407	60.03		
Total	31,110.4	410			

* Predictors: (Constant), culture, EQi-C, WLEIS.

## Data Availability

Analysis script is available on request from the authors.

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
