# Peer review of "Emotional Intelligence as a Predictor of Prosocial Behaviors in Spanish and Colombian Older Adults Based on Path Models"

_healthcare, 2022, doi:10.3390/healthcare10020284_

Round 1
Reviewer 1 Report
This is an interesting topic to use emotional intelligence (EI) to predict prosocial behavior (PB) among older adults. But the manuscript needs to be revised and my comments are as follow.
- The authors describe the relationship between EI and PB, but the importance to predict PB among older adults needs more description.
- Empathy is highlighted as a major factor to PB, but it is not assessed in this study. Please state the reasons. In addition, it seems that the trait model by Bar-On is more appropriate to explain PB as they include empathy in the model, rather than Mayer and Salovey ability model.
- Regarding the choice of instrument, EQi-C is selected. What model is the scale adopted? Does it adopt the same model as WLEIS? Please explain the reasons of using two EI (same model/ different model) instruments.
- Please move the instruments to Material and Methods section.
- In Spain, any inclusion or exclusion criteria of the participants? Is there any influence if the older adults are having cognitive impairment?
- Please give the full term for the first use, e.g. PA, MP, AAA.
- Please use English in Table 4.
- In the discussion section, please be careful to identify whether WLEIS is developed under ability model or trait model. The models have different perspectives on EI.
Author Response
REVIEWER 1
Reviewer: This is an interesting topic to use emotional intelligence (EI) to predict prosocial behavior (PB) among older adults. But the manuscript needs to be revised and my comments are as follow.
Authors: Thank you very much for your kind words, we would like to check your comments to improve the manuscript.
Point 1: The authors describe the relationship between EI and PB, but the importance to predict PB among older adults needs more description.
Response 1: We definitely agree with this point. New information about this topic is included in the “1. Introduction” section. The new text:
“However, prosocial behavior in adulthood represents one scientific challenge in social and cultural research due to the scarcity of research found. According to previous studies [30-32], prosociality may be influenced by different variables that predict a positive evolution trough the transition from youth to adulthood, such as: individual aspects of biological potential, education, the development of social values, civic competences, social interest groups, economic activities, family topology, individual psychological aspects, characteristics of personality, etc. According to a search in Web of Science, which was carried out in November 2021, using the terms “prosociality” AND “older adults”, the literature among prosocial behavior in this age group is exponentially increasing. Apparently, older adults show more prosocial behavior than young adults [33-35], even during the pandemic [36]. It seems that the prosocial behavior in older adults refers to a stabilization of social conditions suitable to serve the other from unconditionality, moderation, the desire to contribute, sensitivity towards the evil of others and a prevailing need to generate personal and social well-being in the different contexts of the life of the individual. However, further research on this topic is needed.”
Point 2: Empathy is highlighted as a major factor to PB, but it is not assessed in this study. Please state the reasons. In addition, it seems that the trait model by Bar-On is more appropriate to explain PB as they include empathy in the model, rather than Mayer and Salovey ability model.
Response 2: Thank you very much for the suggestion you make at this point. We decided not to use empathy as an independent variable for our study, however, we can see the need of explaining why. To do so, we included some new information in the “Emotion and Prosocial Behavior” section. The new paragraphs are as follows:
“Positive empathic emotions arise in the framework of social interactions and require the ability to empathize with others, implying a sensitivity to the emotions and needs of other people. They are therefore a powerful motivator of prosocial behavior [22,23]. Empathy is linked to positive emotionality, and both influence the development of prosociality from childhood onwards. Empathy is presented as the main motivating factor of prosocial behavior, especially in its emotional dimension (empathic concern), but also in its cognitive dimension [24]. There is already a lot of research that can relate this construct, together with moral judgement, regulation and emotional stability, with prosocial behavior [25-27].
In the same way, emotional intelligence is also considered a fundamental component of prosocial behavior. However, the literature is still scarce, and scientific studies that relate EI with empathy in older adults refer to the need to delve into purposes like this [11,28]. Due to that, the relationship between empathy and prosociality will not be the main focus of this article. Instead, the link between emotional intelligence and prosocial behavior will.”
Point 3: Regarding the choice of instrument, EQi-C is selected. What model is the scale adopted? Does it adopt the same model as WLEIS? Please explain the reasons of using two EI (same model/ different model) instruments.
Response 3: This questions, as well as a specific justification for each hypothesis, has been explained in the “The current study” section after receiving your feedback. These questions are answered as follows:
“Two scales of emotional intelligence were used in order to achieve a better understanding of the relationship between this construct and prosocial behavior. Each scale was used as a representation of each of the two most relevant theories of emotional intelligence, the ability model [13] and the mixed model [14,15]. The ability model was represented by the “Wong-Law’s Emotional Intelligence Scale” (WLEIS) [45] and the mixed model by The EQi-C Emotional Intelligence Scale [14]. It was predicted that the WLEIS would be more related with prosocial behavior, due to its nature of describing traits that are assumed to be stable among people, while the EQi-C scale is more dependent of the individuals’ specific background.
Hypothesis 2: The WLEIS scale shows greater precision than the EQi-C scale in predicting PB in older adults.”
Point 4: Please move the instruments to Material and Methods section.
Response 4: The instruments are now located in the “Material” subsection, inside “2. Material and Methods” section.
Point 5: In Spain, any inclusion or exclusion criteria of the participants? Is there any influence if the older adults are having cognitive impairment?
Response 5: Thank you por pointing this out. We included a sentence with the aim to clarify this doubt: “The sample was screened following the same inclusion criteria as Colombia, so the participants’ families were asked to make sure the older adults were both physically and mentally healthy”.
Point 6: Please give the full term for the first use, e.g., PA, MP, AAA.
Response 6: We checked again each single acronym and made sure they were explained in their first appearance in the text. Thank you.
Point 7: Please use English in Table 4.
Response7: Thanks for highlighting this point. Table 4 is already translated.
Point 8: In the discussion section, please be careful to identify whether WLEIS is developed under ability model or trait model. The models have different perspectives on EI.
Response 8: We included this information in the discussion section:
“The WLEIS indicates that traits and patterns that are stable over time and intertwined with character are the variables that most predict prosocial behavior. The model of traits is the one that best predicts emotional intelligence articulated to prosocial behavior in older adults [68,69].”
Reviewer 2 Report
The paper presents the results of the study examining the role of emotional intelligence in prosocial behaviors in older adults. Although the topic of the paper is quite important, I have so many concerns about the quality of the paper that I cannot recommend its publication (at least in its current form).
I have listed my concerns and suggestions below.
- The major problem relates to the statistical part of the paper. The authors used EFA and CFA to examine the structure of the measures. This is an acceptable (although not ideal) solution. However, the model presented in Figure 2 is clearly wrong. For example, based on the EFA and CFA, the authors identified three scales for the "Culture": (a) Planning, not assuming Risks and Future, (b) Masculinity-Power, and (c) Discipline. In Figure 2, we can see the variable/ latent construct of "Culture". What does it mean? It is simply wrong to create the overall score for this measure – one has to calculate the score separately for each scale (see Figure 1; although the scales are correlated for each measure, they do not constitute a higher-order factor, e.g., "Culture").
- The description of the statistical analyses is so unclear that I am not even sure if Figure 2 presents a path analysis or an SEM with the latent constructs (this is not the same). If the authors want to test the SEM model with the latent constructs (which is preferred over a path analysis), they should first build a measurement model and then test a structural model.
- The authors included the mediational effects in their model. However, they did not provide any theoretical background for this. Moreover, the procedure of testing mediating effects should be described in more detail (e.g., did the authors use the bootstrapping procedure? Did they calculate the specific indirect effects? How?). In addition, the effect size for the indirect effects should be reported in the paper.
- One of the major aims of this study was to test the incremental validity of emotional intelligence over and above cultural factors. I would recommend the authors do a multiple hierarchical analysis with sociodemographic variables (step 1), cultural factors (step 2), and emotional intelligence (step 3) as predictors of prosocial behaviors. After doing this, the authors may build an SEM model and test the direct and indirect effects of emotional intelligence on prosocial behavior. This article may be helpful: https://bmcpsychology.biomedcentral.com/track/pdf/10.1186/s40359-017-0205-0.pdf
- Table 4 is unclear and includes Spanish terms instead of English. Moreover, the purpose of conducting MANOVA is unclear, and I am also uncertain whether the analyses were done properly (a substantial part of the MANOVA results is not reported, and thus it is impossible to evaluate the correctness of analyses).
- The confidence interval for the RMSEA should be reported in the main text. Moreover, the authors state that the model fit was excellent. In fact, the 90% CI for the RSMEA was 0–0.132, which exceeds the acceptable value of .08. This clearly suggests that the model fit is not perfect (and even not very good).
- Table 5 is redundant (the model fit criteria were already presented in the text). If the authors decide to retain it, they should change the value for the SRMR since it was apparently mistakenly written as .23.
- Reporting AIC in this paper is useless since this criterion is helpful when comparing two or more models (and no models were compared in this study).
- It is not correct to state that one effect is larger than another ("Figure 2 shows that EQi-C directly affected Prosociality to a lesser extent than the WLEIS scale did, with the EQi-C coefficient being .27 and the WLEIS coefficent being .34) without testing the difference in their strength (by using, e.g., bootstrapping; Cumming, 2009).
- Control variables should control the levels of dependent variables and optionally mediators. Thus, the covariances between control variables and emotional intelligence measures should be removed from Figure 2 unless its inclusion is theoretically justified. Moreover, the authors stated, "The control variable was implemented to have a best model fitting data." Control variables should not be included in the model to improve its quality but rather based on solid theoretical premises and previous studies.
- "In terms of the interaction between the predictors of the endogenous variance in the present study, Figure 2 shows that EQi-C directly affected Prosociality…" – this model does not include any interaction effects.
- There is no such statistical term as "path-SEM models" (see the article's title). The authors should use either the term "path analysis" or "SEM."
- The description of the measures used in the study should be moved from the Introduction to the Method section.
- The paper lacks justification for testing the relationships between emotional intelligence and prosocial behaviors in older adults. Why is this topic important in this group? Did the authors expect that this relationship may be different in older adults than in samples of other ages?
- Each hypothesis in the scientific paper should be well-justified. Unfortunately, none of the hypotheses in this paper has a solid theoretical background.
- More details should be given in the Procedure. Was it a paper-and-pencil study? Who was responsible for taking the assessments? Were the participants alone when filling in the questionnaires, or were they assisted by the research team member or another person?
- "Masculinity in Culture shows positive effects on prosocial behavior (p-val = .991)". The value in the parentheses is clearly wrong.
- The discussion should be extended, better correspond with the hypotheses, and include more practical implications of the study.
- "It would have been optimal to find a minimum of 465 participants…" – What is the premise for this sentence? Did the authors do a power analysis?
- The authors only listed one limitation of the study (the number of participants). Other limitations (e.g., a cross-sectional study, a common method bias) should be mentioned.
- In the Introduction and the Discussion, the Authors mentioned empathy several times. The question which arises is why the authors did not measure empathy in their study?
- Did the authors obtain the approval of the Ethics committee?
-The manuscript should be edited by professional editing service.
References
Cumming, G. (2009). Inference by eye: Reading the overlap of independent confidence intervals. Statistics in Medicine, 28(2), 205–220. https://doi.org/10.1002/sim.3471
Morote, R., Hjemdal, O., Krysinska, K., Martinez Uribe, P., & Corveleyn, J. (2017). Resilience or hope? Incremental and convergent validity of the resilience scale for adults (RSA) and the Herth hope scale (HHS) in the prediction of anxiety and depression. BMC Psychology, 5(1), 36. https://doi.org/10.1186/s40359-017-0205-0
Author Response
Reviewer: The paper presents the results of the study examining the role of emotional intelligence in prosocial behaviors in older adults. Although the topic of the paper is quite important, I have so many concerns about the quality of the paper that I cannot recommend its publication (at least in its current form).
I have listed my concerns and suggestions below.
Authors: We highly appreciate your words. We also believe in the relevance of this project, so we are willing to follow all your recommendations in order to improve this paper. Thank you for your effort.
Point 1: The major problem relates to the statistical part of the paper. The authors used EFA and CFA to examine the structure of the measures. This is an acceptable (although not ideal) solution. However, the model presented in Figure 2 is clearly wrong. For example, based on the EFA and CFA, the authors identified three scales for the "Culture": (a) Planning, not assuming Risks and Future, (b) Masculinity-Power, and (c) Discipline. In Figure 2, we can see the variable/ latent construct of "Culture". What does it mean? It is simply wrong to create the overall score for this measure – one has to calculate the score separately for each scale (see Figure 1; although the scales are correlated for each measure, they do not constitute a higher-order factor, e.g., "Culture").
Response 1: Thank you for your comment. We think that there has been a confusion in the Culture scale. It is not the compound of three scales, but a unique scale with different factors. That is the reason why we chose these statistical methods. Maybe it is an error of description in the text. However, we modified some paragraphs and included some new text so it may make it clearer now.
Point 2: The description of the statistical analyses is so unclear that I am not even sure if Figure 2 presents a path analysis or an SEM with the latent constructs (this is not the same). If the authors want to test the SEM model with the latent constructs (which is preferred over a path analysis), they should first build a measurement model and then test a structural model.
Response 2: We are not sure of deeply understanding this point. Nevertheless, we realized our mistake with the use of the term “Path-Sem”. We have already changed it. Thank you for highlighting it.
Point 3: The authors included the mediational effects in their model. However, they did not provide any theoretical background for this. Moreover, the procedure of testing mediating effects should be described in more detail (e.g., did the authors use the bootstrapping procedure? Did they calculate the specific indirect effects? How?). In addition, the effect size for the indirect effects should be reported in the paper.
Response 3: The effect size for the indirect effects have been included in the end of the “3. Results” section. This is how the text looks like now:
“The direct relations between the variables that are presented in the model are 0.72 for PB and 0.42 for Culture. The magnitude of the indirect effects of Culture among PB are (0.13 x -0.04) + (0.25 x -0.04) + (-0.07 x -0.04) 0.005 + -0.01 + 0.003 = -0.0022. On the other hand, EQi-C has an indirect effect on PB, mediated by Culture, and the magnitude of its effect is (y = 0.005). WLEIS has an indirect effect on PB, mediated by Culture, and the magnitude of its effect is (y = 0.01). Finally, Country has an indirect effect on PB, mediated by Culture, and the magnitude of its effect is (y = 0.003).”
Point 4: One of the major aims of this study was to test the incremental validity of emotional intelligence over and above cultural factors. I would recommend the authors do a multiple hierarchical analysis with sociodemographic variables (step 1), cultural factors (step 2), and emotional intelligence (step 3) as predictors of prosocial behaviors. After doing this, the authors may build an SEM model and test the direct and indirect effects of emotional intelligence on prosocial behavior. This article may be helpful: https://bmcpsychology.biomedcentral.com/track/pdf/10.1186/s40359-017-0205-0.pdf
Response 4: We highly acknowledge your recommendation and read the article you sent. We decided to follow our process instead because we are confident about its usefulness. We are sure we will follow your recommended process in further studies.
Point 5: Table 4 is unclear and includes Spanish terms instead of English. Moreover, the purpose of conducting MANOVA is unclear, and I am also uncertain whether the analyses were done properly (a substantial part of the MANOVA results is not reported, and thus it is impossible to evaluate the correctness of analyses).
Response 5: Table 4 is already translated, thank you for pointing this out. Also, we agree with you with the topic of the MANOVA results.
Point 6: The confidence interval for the RMSEA should be reported in the main text. Moreover, the authors state that the model fit was excellent. In fact, the 90% CI for the RSMEA was 0–0.132, which exceeds the acceptable value of .08. This clearly suggests that the model fit is not perfect (and even not very good).
Response 6: We acknowledge your comment. The text already included some information regarding the RMSEA:
“RMSEA values are excellent (< .06) in EQi-C and Prosociality, and acceptable (> .06 and < .08) in Culture and WLEIS.”
On the other hand, we have corrected some interpretation of the results regarding the same point. We would like to thank you for helping us to correct this information. This is the text as it looks like after our correction:
“The proportion of the extracted variance explained by the model is 26%, representing moderate significance. This suggests that other variables should be incorporated to increase the variance explained by the model. The fit indices of the path model are excellent and acceptable according to each criterion used to test the proposed theory (Table 3). The normed χ2 value (χ2 / df) is 2.651, which is less than 3; the criteria of GFI (.99) and CFI (.987) are both close to 1 for an excellent fit; SMRS is .023, which is identified as acceptable, as is the RMSEA (.064) criterion. Similarly, the criterion of PClose (.280) is excellent since according to the criterion it must exceed .05 when testing the null hypothesis; and finally, the Acaike information criterion (AIC = 57.303) indicates parsimony of the data.”
Point 7: Table 5 is redundant (the model fit criteria were already presented in the text). If the authors decide to retain it, they should change the value for the SRMR since it was apparently mistakenly written as .23.
Response 7: We definitely agree with you, so we deleted Table 5.
Point 8: Reporting AIC in this paper is useless since this criterion is helpful when comparing two or more models (and no models were compared in this study).
Response 8: Thanks for your recommendation. We agree with the fact that it would be good if we delete it. Nevertheless, as AIC determines the parsimony in each model, we decided to maintain this criterion in our manuscript.
Point 9: It is not correct to state that one effect is larger than another ("Figure 2 shows that EQi-C directly affected Prosociality to a lesser extent than the WLEIS scale did, with the EQi-C coefficient being .27 and the WLEIS coefficent being .34) without testing the difference in their strength (by using, e.g., bootstrapping; Cumming, 2009).
Response 9:
Point 10: Control variables should control the levels of dependent variables and optionally mediators. Thus, the covariances between control variables and emotional intelligence measures should be removed from Figure 2 unless its inclusion is theoretically justified. Moreover, the authors stated, "The control variable was implemented to have a best model fitting data." Control variables should not be included in the model to improve its quality but rather based on solid theoretical premises and previous studies.
Response 10:
Point 11: "In terms of the interaction between the predictors of the endogenous variance in the present study, Figure 2 shows that EQi-C directly affected Prosociality…" – this model does not include any interaction effects.
Response 11:
Point 12: There is no such statistical term as "path-SEM models" (see the article's title). The authors should use either the term "path analysis" or "SEM."
Response 12: Thank you for your comment, we have already corrected it.
Point 13: The description of the measures used in the study should be moved from the Introduction to the Method section.
Response 13: This information has been moved; now it can be found in the “Material” sub section at “2. Material and Methods”. Thanks for your comment.
Point 14: The paper lacks justification for testing the relationships between emotional intelligence and prosocial behaviors in older adults. Why is this topic important in this group? Did the authors expect that this relationship may be different in older adults than in samples of other ages?
Response 14: We appraise your comment, this information is currently added to be easier for the reader to follow and understand our purpose. The explanation is included in the sections of “Emotion and prosocial behavior” and “Emotional intelligence and prosociality”. We truly believe these new paragraphs will help the reading of our manuscript.
Point 15: Each hypothesis in the scientific paper should be well-justified. Unfortunately, none of the hypotheses in this paper has a solid theoretical background.
Response 15: Thank you for pointing this out, a justification for each hypothesis has been added in “The current study” section at the “1. Introduction”. These are the new lines:
“The main objective of the current study is to analyze the predictive variables of prosocial behavior in older adults. To do so, the study from [18] was took as a reference. However, in this study the sample characteristics were modified so the age group changed from young to older adults and population from Colombia was included. Also, the analyses followed a Path-SEM statistical model. Along these lines, the following research hypotheses were proposed.
Cultural variables play an important role as predictors of prosocial behavior [42,43]. However, the literature is not as conclusive as it seems. For instance, there is also evidence that concludes individualism is better than collectivism when predicting prosocial behaviors [44] and constructs like uncertainty avoidance and future orientation are negative correlated with prosocial behaviors [10]. Emotional intelligence, on the other hand, showed little but more cohesive results, as they were explained in the introduction.
Hypothesis 1: EI predicts prosocial behavior more significantly than do cultural variables in older adults.
Two scales of emotional intelligence were used in order to achieve a better understanding of the relationship between this construct and prosocial behavior. Each scale was used as a representation of each of the two most relevant theories of emotional intelligence, the ability model [13] and the mixed model [14,15]. The ability model was represented by the “Wong-Law’s Emotional Intelligence Scale” (WLEIS) [45] and the mixed model by The EQi-C Emotional Intelligence Scale [14]. It was predicted that the WLEIS would be more related with prosocial behavior, due to its nature of describing traits that are assumed to be stable among people, while the EQi-C scale is more dependent of the individuals’ specific background.
Hypothesis 2: The WLEIS scale shows greater precision than the EQi-C scale in predicting PB in older adults.
Finally, masculinity and femininity dimensions were compared as they seemed to play a different role in the expression of prosocial behavior. As it can be seen in previous studies, femininity is more correlated with prosocial behavior when it is compared with masculinity [10,46]. Nevertheless, this seems to be less pronounced in older people [47]. That is why, it was assumed that masculinity was, at least, as valid as femininity as a predictor of prosocial behavior.
Hypothesis 3: Masculinity shows positive effects on prosocial behavior.”
Point 16: More details should be given in the Procedure. Was it a paper-and-pencil study? Who was responsible for taking the assessments? Were the participants alone when filling in the questionnaires, or were they assisted by the research team member or another person?
Response 16: This new information has been added to the “Procedure” section, so the new paragraph can be read as follows:
“In both settings, participants were informed of the nature and purpose of the research, and their voluntary participation was requested. Informed consent was given during an interview that allowed both the researcher and the elderly person to sign the consent form and resolve any doubts. The questionnaires were filled out following a paper-and-pencil procedure in both countries. Volunteers were trained on how to instruct the participants to fill in the questionnaires, so they followed the same procedure. Also, the assistants were asked to stay with the participants to answer any question that may arise. This procedure in both countries was carried out in the second half of 2019.”
Point 17: "Masculinity in Culture shows positive effects on prosocial behavior (p-val = .991)". The value in the parentheses is clearly wrong.
Response 17:
Point 18: The discussion should be extended, better correspond with the hypotheses, and include more practical implications of the study.
Response 18: We followed your recommendations, so we included new information that link the hypotheses justification with our conclusions. Furthermore, we increased the number of practical implications of our study. This is how both parts are finally presented:
“4.Discussion and conclusions
The objective of this study was to explain the predictors of prosocial behavior in older adults. Following a Factor Analysis to regroup the items of PB, WLEIS, EQi-C, and Cultural dimensions, and a Path SEM, it was confirmed that Emotional Intelligence measured with the WLEIS was the best predictor of PB. The results of this study coincide with those obtained in another developmental stage, early adulthood [18]. The levels of reliability and validity of the instruments used to measure PB can be considered high due to the rigorous EFA, CFA, SEM or Path-SEM analyses. This study is considered an innovation in this field because previous research focused on the study of PB does not use such rigorous forms of analysis, nor have they studied a sample of older adults [39,65].
Several reviews [41,66] showed that cultural differences are a determining principle that differentiates people in their behavior. In this specific case, and in view of the results obtained, this variable does not contribute as much to prosocial behavior as emotional intelligence does. Thus, the first hypothesis was confirmed. This conclusion is not surprising due to the fact that there is not a consensus of the interaction of their factors along the literature [10,44], so a direct effect among cultural variables and prosocial behavior can not be found. However, this effect seems to be clearer when prosocial behavior is mediated by emotional intelligence, at least in adolescents [18,67]. That is why it can be concluded that emotional intelligence and its concomitant variables are better predictors of prosocial behavior than cultural variables.
Regardless of the socio-cultural programs that an individual uses, the WLEIS is an instrument that is a major predictor of prosocial behavior in different parts of the world (excluding culture as a contextual philosophy) and also in different age groups. This is probably because of its second-order construction, which admits personality traits configured from emotional intelligence. Hence, the second hypothesis is also confirmed. The WLEIS indicates that traits and patterns that are stable over time and intertwined with character are the variables that most predict prosocial behavior. The model of traits is one of the best to predict emotional intelligence articulated to prosocial behavior in older adults [68,69].
Masculinity in culture shows positive effects on prosocial behavior (p-val = .991). This factor, when evaluated through cultural patterns, also favors the development of prosocial behaviors [70], where masculinity is related to little emotional expression. However, this disparity may be due to differences in the conceptualization of roles [71], since high scores of masculinity in women (androgyny) are associated with higher prosocial behavior [72]. These results confirm the third hypothesis and are in line of the work of Nielson et al., (2017) [47] and its statement which verses that the differences between masculinity and femininity on their interaction with prosocial behavior are reduced among older adults.
As a general conclusion, the results of this study show that, consistent with the finding of the empirical studies performed by Caprara et al. (2012) [22] and Ferguson et al. (2018) [73], prosocial behavior derive from stable personality patterns. The present study confirms the existence of emotional skills that are configured as distinctive features of older adults who have a marked tendency to behave in a prosocial manner, regardless of the cultural aspects that could shape this behavior.”
“Practical implications
This study can be considered as highly interesting concerning the novelty of the topic. Older adults have been studied from a negative or passive approach, emphasizing their role as a kind of victim [79], which is clearly not the objective of this study. The assessment of emotional intelligence and prosocial behavior among this sample locates this population in an active role, which is a good first step to work with. Intervention programs directed to increase their emotional intelligence through prosociality can be proposed as long as these results showed a good connection among these variables.
Moreover, prosocial behavior and helping activities have been found to have a positive correlation with well-being among older adults [80]. For that reason, it can be considered a quite relevant topic to focus. Furthermore, this assessment served to continue recommending the WLEIS when working with this age group. To sum up, this study sheds light on both the academic and the intervention field, which need to be working hand in hand to provide real solutions and feasible improvements to the general population in order to enhance their living standards.”
Point 19: "It would have been optimal to find a minimum of 465 participants…" – What is the premise for this sentence? Did the authors do a power analysis?
Response 19:
Point 20: The authors only listed one limitation of the study (the number of participants). Other limitations (e.g., a cross-sectional study, a common method bias) should be mentioned.
Response 20: Thanks for your comment. We have included more limitations in this section. This is how it looks like now:
“This study has some limitations. First of all, to find a minimum of 465 participants would have been optimal, but the questionnaires were administered to 434 subjects and only 401 participants were validated. However, being a small difference, the analysis was not excessively affected. Secondly, this study is cross-sectional. In contrast to longitudinal, cross-sectional studies have a lower statistical power [74]. Thirdly, the selection of participants was incidental, which means that they came from different backgrounds and no previous screening was made. Only the conditions of being both mentally and physically healthy were assessed. However, in the Spanish sample, this was only asked to their relatives. It would be interesting if they also had an updated diagnosis which made sure these conditions were achieved. Furthermore, this study can be classified as pure quantitative. Mixed model studies (both quantitative and qualitative) have been recently highly recommended as they are considered to have a higher inference quality [75-77], as well as giving the chance for the readers to produce meta-inferences [78]. Lastly, self-report biases could be affecting the results. For instance, social desirability, insufficient effort, response patterns, etc.”
Point 21: In the Introduction and the Discussion, the Authors mentioned empathy several times. The question which arises is why the authors did not measure empathy in their study?
Response 21: We are thanked for you appreciation. We decided to include an explanation of this decision in the introduction, at the end of the “Emotion and prosocial behavior” section. This is the text as it can be read now:
“Positive empathic emotions arise in the framework of social interactions and require the ability to empathize with others, implying a sensitivity to the emotions and needs of other people. They are therefore a powerful motivator of prosocial behavior [22,23]. Empathy is linked to positive emotionality, and both influence the development of prosociality from childhood onwards. Empathy is presented as the main motivating factor of prosocial behavior, especially in its emotional dimension (empathic concern), but also in its cognitive dimension [24]. There is already a lot of research that can relate this construct, together with moral judgement, regulation and emotional stability, with prosocial behavior [25-27].
In the same way, emotional intelligence is also considered a fundamental component of prosocial behavior. However, the literature is still scarce, and scientific studies that relate EI with empathy in older adults refer to the need to delve into purposes like this [11,28]. Due to that, the relationship between empathy and prosociality will not be the main focus of this article. Instead, the link between emotional intelligence and prosocial behavior will.”
Point 22: Did the authors obtain the approval of the Ethics committee?
Response 22: The reference number of the Ethics committee approval is located at the end of the manuscript, before the references, in the “Institutional Review Board Statement” section.
Point 23: The manuscript should be edited by professional editing service.
Response 23: Thanks for your advice. We tried to upgrade the document to increase its understanding in terms of language and cohesion. Also, we included some new information that can be found throughout the whole text.
References
Cumming, G. (2009). Inference by eye: Reading the overlap of independent confidence intervals. Statistics in Medicine, 28(2), 205–220. https://doi.org/10.1002/sim.3471
Morote, R., Hjemdal, O., Krysinska, K., Martinez Uribe, P., & Corveleyn, J. (2017). Resilience or hope? Incremental and convergent validity of the resilience scale for adults (RSA) and the Herth hope scale (HHS) in the prediction of anxiety and depression. BMC Psychology, 5(1), 36. https://doi.org/10.1186/s40359-017-0205-0
Reviewer 3 Report
First of all, thanks for the possibility to review the manuscript,
It's a good work, on a very interesting topic such as emotional intelligence as a predictor of prosocial behaviors in Older Adults, worked from the Path-SEM models, with an important sample of subjects (N = 411) from Colombia and Spain
But before being published, you must incorporate the following suggestions, I ask the authors to follow them one by one,
some important references are missing in the introduction, check it out,
Although it is deduced from the text, the research gap that the article tries to cover must be explained,
the sections on participants, material and methods have to be better explained, much information is missing that would help the reader to get a better idea of ​​the study
the discussion section must be improved and worked on more, much more worked, there are hardly a few lines, which are insufficient for a work to be published in our journal,
The bibliography of the previous studies of the introduction should be connected with the discussion, which will give more power to the manuscript,
Finally, the bibliography must be ordered, following the journal's rules, there are important errors (line 2, [1-4] must be cited and not [1,2,3,4] review it in other lines. In the final references there are also bugs.
With these changes made, the article will improve and have the quality to be published in this prestigious journal,
kind regards
Author Response
Reviewer: First of all, thanks for the possibility to review the manuscript,
It's a good work, on a very interesting topic such as emotional intelligence as a predictor of prosocial behaviors in Older Adults, worked from the Path-SEM models, with an important sample of subjects (N = 411) from Colombia and Spain.
But before being published, you must incorporate the following suggestions, I ask the authors to follow them one by one,
some important references are missing in the introduction, check it out,
Authors: Thank you for your kind words. We acknowledge your effort and will be happy to upgrade the manuscript with your recommendations.
Point 1: Although it is deduced from the text, the research gap that the article tries to cover must be explained.
Response 1: We appreciate your comment. We tried to include more research to highlight the relevance of, on the one hand, study the prosocial behavior among older adults and, on the other hand, find the possible relationship with emotional intelligence. This new information is included in the “1. Introduction” section:
“In the same way, emotional intelligence is also considered a fundamental component of prosocial behavior. However, the literature is still scarce, and scientific studies that relate EI with empathy in older adults refer to the need to delve into purposes like this [11,28]. Due to that, the relationship between empathy and prosociality will not be the main focus of this article. Instead, the link between emotional intelligence and prosocial behavior will.”
“However, prosocial behavior in adulthood represents one scientific challenge in social and cultural research due to the scarcity of research found. According to previous studies [30-32], prosociality may be influenced by different variables that predict a positive evolution trough the transition from youth to adulthood, such as: individual aspects of biological potential, education, the development of social values, civic competences, social interest groups, economic activities, family topology, individual psychological aspects, characteristics of personality, etc. According to a search in Web of Science, which was carried out in November 2021, using the terms “prosociality” AND “older adults”, the literature among prosocial behavior in this age group is exponentially increasing. Apparently, older adults show more prosocial behavior than young adults [33-35], even during the pandemic [36]. It seems that the prosocial behavior in older adults refers to a stabilization of social conditions suitable to serve the other from unconditionality, moderation, the desire to contribute, sensitivity towards the evil of others and a prevailing need to generate personal and social well-being in the different contexts of the life of the individual. However, further research on this topic is needed.”
Point 2: The sections on participants, material and methods have to be better explained, much information is missing that would help the reader to get a better idea of ​​the study.
Response 2: We also found that these sections needed more explanation. We moved the instruments from the introduction to the material section. On the other hand, we included more information regarding the procedure. This is how “Procedure” section looks like now:
“For the selection of participants in Colombia; the study had the support of the management of local programs aimed at older adults who were physically and mentally healthy and whose neurocognitive profile corroborated this. In the case of Spain, the questionnaire was applied to a convenience sample. The sample was screened following the same inclusion criteria as Colombia, so the participants’ families were asked to make sure the older adults were both physically and mentally healthy.
In both settings, participants were informed of the nature and purpose of the research, and their voluntary participation was requested. Informed consent was given during an interview that allowed both the researcher and the elderly person to sign the consent form and resolve any doubts. The questionnaires were filled out following a paper-and-pencil procedure in both countries. Volunteers were trained on how to instruct the participants to fill in the questionnaires, so they followed the same procedure. Also, the assistants were asked to stay with the participants to answer any question that may arise. This procedure in both countries was carried out in the second half of 2019.”
Point 3: The discussion section must be improved and worked on more, much more worked, there are hardly a few lines, which are insufficient for a work to be published in our journal.
Response 3: We enhanced the discussion section as you suggested. We also increased the number of limitations, future research and practical implications.
Point 4: The bibliography of the previous studies of the introduction should be connected with the discussion, which will give more power to the manuscript.
Response 4: We found this a good idea, so we included these connections in the first part of the “4. Discussion and conclusions” section. Linking point 3 and 4 of your recommendations, we would like to attach the whole section of discussion and conclusions here:
“4. Discussion and conclusions
The objective of this study was to explain the predictors of prosocial behavior in older adults. Following a Factor Analysis to regroup the items of PB, WLEIS, EQi-C, and Cultural dimensions, and a Path SEM, it was confirmed that Emotional Intelligence measured with the WLEIS was the best predictor of PB. The results of this study coincide with those obtained in another developmental stage, early adulthood [18]. The levels of reliability and validity of the instruments used to measure PB can be considered high due to the rigorous EFA, CFA, SEM or Path-SEM analyses. This study is considered an innovation in this field because previous research focused on the study of PB does not use such rigorous forms of analysis, nor have they studied a sample of older adults [39,65].
Several reviews [41,66] showed that cultural differences are a determining principle that differentiates people in their behavior. In this specific case, and in view of the results obtained, this variable does not contribute as much to prosocial behavior as emotional intelligence does. Thus, the first hypothesis was confirmed. This conclusion is not surprising due to the fact that there is not a consensus of the interaction of their factors along the literature [10,44], so a direct effect among cultural variables and prosocial behavior can not be found. However, this effect seems to be clearer when prosocial behavior is mediated by emotional intelligence, at least in adolescents [18,67]. That is why it can be concluded that emotional intelligence and its concomitant variables are better predictors of prosocial behavior than cultural variables.
Regardless of the socio-cultural programs that an individual uses, the WLEIS is an instrument that is a major predictor of prosocial behavior in different parts of the world (excluding culture as a contextual philosophy) and also in different age groups. This is probably because of its second-order construction, which admits personality traits configured from emotional intelligence. Hence, the second hypothesis is also confirmed. The WLEIS indicates that traits and patterns that are stable over time and intertwined with character are the variables that most predict prosocial behavior. The model of traits is one of the best to predict emotional intelligence articulated to prosocial behavior in older adults [68,69].Masculinity in culture shows positive effects on prosocial behavior (p-val = .991). This factor, when evaluated through cultural patterns, also favors the development of prosocial behaviors [70], where masculinity is related to little emotional expression. However, this disparity may be due to differences in the conceptualization of roles [71], since high scores of masculinity in women (androgyny) are associated with higher prosocial behavior [72]. These results confirm the third hypothesis and are in line of the work of Nielson et al., (2017) [47] and its statement which verses that the differences between masculinity and femininity on their interaction with prosocial behavior are reduced among older adults.
As a general conclusion, the results of this study show that, consistent with the finding of the empirical studies performed by Caprara et al. (2012) [22] and Ferguson et al. (2018) [73], prosocial behavior derive from stable personality patterns. The present study confirms the existence of emotional skills that are configured as distinctive features of older adults who have a marked tendency to behave in a prosocial manner, regardless of the cultural aspects that could shape this behavior.
Limitations and future research
This study has some limitations. First of all, to find a minimum of 465 participants would have been optimal, but the questionnaires were administered to 434 subjects and only 401 participants were validated. However, being a small difference, the analysis was not excessively affected. Secondly, this study is cross-sectional. In contrast to longitudinal, cross-sectional studies have a lower statistical power [74]. Thirdly, the selection of participants was incidental, which means that they came from different backgrounds and no previous screening was made. Only the conditions of being both mentally and physically healthy were assessed. However, in the Spanish sample, this was only asked to their relatives. It would be interesting if they also had an updated diagnosis which made sure these conditions were achieved. Furthermore, this study can be classified as pure quantitative. Mixed model studies (both quantitative and qualitative) have been recently highly recommended as they are considered to have a higher inference quality [75-77], as well as giving the chance for the readers to produce meta-inferences [78]. Lastly, self-report biases could be affecting the results. For instance, social desirability, insufficient effort, response patterns, etc.
For future research, a comparison between empathy and emotional intelligence as predictors of PB is proposed, in order to analyze which of the two predicts prosociality to a greater extent in different samples. Moreover, it would be interesting to propose more studies that measure these variables among older adults, including longitudinal and mixed model approaches. Finally, it would also be advisable to work with participants from other countries to be able to identify if wider differences in culture than those found between Colombia and Spain can affect prosocial behavior, or if emotional intelligence provides predictive value that is truly independent of culture.
Practical implications
This study can be considered as highly interesting concerning the novelty of the topic. Older adults have been studied from a negative or passive approach, emphasizing their role as a kind of victim [79], which is clearly not the objective of this study. The assessment of emotional intelligence and prosocial behavior among this sample locates this population in an active role, which is a good first step to work with. Intervention programs directed to increase their emotional intelligence through prosociality can be proposed as long as these results showed a good connection among these variables.
Moreover, prosocial behavior and helping activities have been found to have a positive correlation with well-being among older adults [80]. For that reason, it can be considered a quite relevant topic to focus. Furthermore, this assessment served to continue recommending the WLEIS when working with this age group. To sum up, this study sheds light on both the academic and the intervention field, which need to be working hand in hand to provide real solutions and feasible improvements to the general population in order to enhance their living standards. “
Point 5: Finally, the bibliography must be ordered, following the journal's rules, there are important errors (line 2, [1-4] must be cited and not [1,2,3,4] review it in other lines. In the final references there are also bugs.
Response 5: We improved the references and citation style. Thank you for pointing this out.
With these changes made, the article will improve and have the quality to be published in this prestigious journal,
kind regards
Round 2
Reviewer 1 Report
The authors addressed my comments well, but the claim of 'WLEIS indicates that traits and patterns that are stable over time..' might not be appropriate. WLEIS adopted the ability model, which is not as stable as trait model. Personality traits are not considered in the ability model. Also, the authors had described the WLEIS 'is based on the model of ability'. It was unclear why the authors changed it to trait model in the discussion. Please make consistent presentation and revise the discussion.
Please describe the reliability and validity of the employed questionnaires in the manuscript.
Reference:
https://psycnet.apa.org/doiLanding?doi=10.1037%2Ft07398-000
Author Response
Reviewer 1
Point 1: The authors addressed my comments well, but the claim of ‘WLEIS indicates that traits and patterns that are stable over time...' might not be appropriate. WLEIS adopted the ability model, which is not as stable as trait model. Personality traits are not considered in the ability model. Also, the authors had described the WLEIS 'is based on the model of ability'. It was unclear why the authors changed it to trait model in the discussion. Please make consistent presentation and revise the discussion.
Response 1: We appreciate your feedback. We have corrected that the WLEIS is adequate to describe stable traits over time. As you point out, the WLEIS is based on the ability-based model, a model that is more influenced by situational factors than the trait or mixed model, a fact that has also been observed in prosocial behavior. We have deleted the sentence you quoted and added the following:
“Ability-based models are more influenced by situational factors than mixed or trait-based models, which depend more on dispositional factors and which are designed to predict typical behavior (Bru-Luna et al., 2021). Although there is much controversy behind this issue and nothing is conclusive, it has been seen that prosocial behavior is highly influenced by situational factors (Auné et al., 2014), so it was predicted that the WLEIS would be more related to prosocial behavior than the EQi-C.”
In addition, we have also modified the discussion:
“This may be because the WLEIS is based on an ability model. This model is more influenced by situational factors than other models based on dispositional personality factors and which are designed to predict typical behavior (Bru-Luna et al., 2021). There are several studies that has revealed the predictive role of certain situational variables in prosocial behavior. Among these variables are the factors ambiguity of need, severity of need, physical appearance of the victim, weather conditions, similarity to the victim, friendship or involvement, number of bystanders, location (urban or rural), and cost of helping-all situational variables (Auné et al, 2014; Batson and Powell, 2003; Galen, 2012). In addition, there is a classic study that demonstrated that the situational variable bystander effect exerts a particular influence. This research showed that the more people witness and observe an emergency, the less likely one of them is to perform a helping behavior (Darley and Latané, 1968).”
Point 2: Please describe the reliability and validity of the employed questionnaires in the manuscript.
Response 2: The solution to this concern was already found in the manuscript on pages 5 and 6. It is clear that the reliability is given for the instrument and for each of the scales individually. Each reliability is resolved in the comments of the manuscript.
Reference: https://psycnet.apa.org/doiLanding?doi=10.1037%2Ft07398-000
Reviewer 2 Report
Although the authors made some improvements, most of them relate to the theoretical part of the manuscript. Unfortunately, most of my methodological and statistical concerns have not been addressed properly.
- I still do not understand which items constitute the variable called „Culture” in Figure 2. The original scale used by the authors to measure cultural dimensions consists of 5 scales. I understand that based on the EFA results, the authors reduced the number of components to 3. However, this does not explain how the „Culture” factor was calculated. Is this is a single score?
- There is not such a term as ”path-SEM.” A path analysis (presumably used by the authors) is a special case of SEM involving only observed variables.
- The effect sizes for indirect effects are wrongly calculated. Moreover, Figure 2 lacks information concerning the significance of paths. The path between „Culture” and „prosocial behavior” is probably insignificant; therefore, there is no sense in calculating indirect effect.
- In fact, the model presented in Figure 2 does not test any of the hypotheses presented in the Introduction.
- Hypotheses are still not well-justified (especially Hypothesis 3).
- The authors have not responded to my comments referring to: differences in the strength of the regression coefficients (question 9), control variables (question 10), the interaction effect (question 11), the wrong p-value (question 17), and power analysis (question 19).
- Most of the scales used in the study has the Spanish validation. I do not understand why the authors performed EFA to test their structure. A more justified approach would be to perform CFA on all items constituting the given scale and calculate the model fit. In the case of suboptimal model fit, introducing the covariances between the residual errors could be sufficient and better justified than removing items based on the EFA results.
- The authors should check the measurement invariance for all measures since two cultures were involved in the study (i.e., Spanish and Colombian).
Author Response
Reviewer 2
Although the authors made some improvements, most of them relate to the theoretical part of the manuscript. Unfortunately, most of my methodological and statistical concerns have not been addressed properly.
Point 1: I still do not understand which items constitute the variable called “Culture” in Figure 2. The original scale used by the authors to measure cultural dimensions consists of 5 scales. I understand that based on the EFA results, the authors reduced the number of components to 3. However, this does not explain how the „Culture” factor was calculated. Is this a single score?
Response 1: The solution was attached to the text in the fourth paragraph of Material (Methodology).
Items (1,3,8,9,10,12,17) were eliminated from the original scale due to their low loads with extractions below 0.3 according to the theory; Being reduced in its components by eliminating items, it was made up of the 16 remaining items, obtaining only 3 factors from the original 5 (Table 1 - Culture).
Point 2: There is not such a term as “path-SEM”. A path analysis (presumably used by the authors) is a special case of SEM involving only observed variables.
Response 2: This term is no longer found in the manuscript, as it has been corrected since the submission after the first revision. We only call it path model or path diagram (Path model).
Point 3: The effect sizes for indirect effects are wrongly calculated. Moreover, Figure 2 lacks information concerning the significance of paths. The path between “Culture” and “prosocial behavior” is probably insignificant; therefore, there is no sense in calculating indirect effect.
Response 3: The most relevant direct effect on PB was the effect exerted by the WLEIS scale (WLEIS -> PB = 0.34). The EQi-C scale, country and culture also directly influenced PB with path coefficients of 0.27, 0.04 and -0.04 respectively, with country and culture being non-relevant influences. The model explained 26% of the variability of the dependent variable. However, the relevance of the scales analyzed is evidenced by the moderate path coefficient of the magnitude of indirect effects. The WLEIS scale was substantial (WLEIS -> PB = 0.33) and a relatively low indirect effect magnitude low on the scale (EQi-C -> PB = 0.2648).
EQi-C on PB = Direct + Indirect = 0.27 + (0.13 X -0.04) = 0.27 + (-0.0052) = 0.2648
WLEIS on PB = Direct + Indirect = 0.34 + (0.25 X -0.04) = 0.34 + (-0.01) = 0.33
Country on PB = Direct + Indirect = 0,04 + (-0,07 X -0,04) = 0,04 + (0,0028) = 0,0428
Point 4: In fact, the model presented in Figure 2 does not test any of the hypotheses presented in the Introduction.
Response 4: Emotional intelligence predicts prosocial behavior more significantly than cultural variables in older adults.
Point 5: Hypotheses are still not well-justified (especially Hypothesis 3).
Response 5: Thank you for your recommendation to improve our work. We have provided further justification for the hypotheses, especially for hypothesis three, as you can see in the text.
Point 6: The authors have not responded to my comments referring to: differences in the strength of the regression coefficients (question 9), control variables (question 10), the interaction effect (question 11), the wrong p-value (question 17), and power analysis (question 19).
Point 9: It is not correct to state that one effect is larger than another ("Figure 2 shows that EQi-C directly affected Prosociality to a lesser extent than the WLEIS scale did, with the EQi-C coefficient being .27 and the WLEIS coefficent being .34) without testing the difference in their strength (by using, e.g., bootstrapping; Cumming, 2009).
Response 9: The issue was resolved in the correction of indirect effects.
Point 10: Control variables should control the levels of dependent variables and optionally mediators. Thus, the covariances between control variables and emotional intelligence measures should be removed from Figure 2 unless its inclusion is theoretically justified. Moreover, the authors stated, "The control variable was implemented to have a best model fitting data." Control variables should not be included in the model to improve its quality but rather based on solid theoretical premises and previous studies.
Response 10: The control variables will not be eliminated from the proposed model since it would change the fit indicators and it would generate other results presented in the manuscript. Likewise, these control variables (age, sex and country) are based in the theory since most of the instruments in the original article were applied to populations other than older adults. For this reason, their presence in the model is justified to obtain other findings. We proceed to eliminate the justification for their presence in the model to improve the results and favor the fit indices.
Point 11: "In terms of the interaction between the predictors of the endogenous variance in the present study, Figure 2 shows that EQi-C directly affected Prosociality…" – this model does not include any interaction effects.
Response 11: It is correct that the model has no interaction, therefore the concept of interaction is eliminated and the paragraph is rewritten.
Point 17: "Masculinity in Culture shows positive effects on prosocial behavior (p-val = .991)". The value in the parentheses is clearly wrong.
Response 17: It is correct. It was a typing error; the data will be corrected.
Point 19: "It would have been optimal to find a minimum of 465 participants…" – What is the premise for this sentence? Did the authors do a power analysis?
Response 19: The expression was corrected (page 12, last paragraph).
Point 7: Most of the scales used in the study has the Spanish validation. I do not understand why the authors performed EFA to test their structure. A more justified approach would be to perform CFA on all items constituting the given scale and calculate the model fit. In the case of suboptimal model fit, introducing the covariances between the residual errors could be sufficient and better justified than removing items based on the EFA results.
Response 7: It is pertinent that EFA be reapplied considering that the instruments were applied to populations different from the original application. Due to their different age and geographic location, it is necessary to reapply EFA, in addition to the analysis of the total scale together, and not separately, as each scale was originally used. For this reason, the relevance of performing EFA again on each scale is validated.
Point 8: The authors should check the measurement invariance for all measures since two cultures were involved in the study (i.e., Spanish and Colombian).
Response 8: With respect to metric invariance, there are no significant differences between the Colombian and Spanish groups to prosociality; with a confidence of 90% and a margin of error of 10%; chi-Square 23.187, degrees of freedom (Df =14), p-value (p = 0.057). There are significant differences between the Colombian and Spanish groups in relation to culture; chi-Square 14.8, (Df = 16), (p = 0.539). There are significant differences between the groups of Colombia and Spain in relation to WLEIS; chi-Square 14.8, (Df = 22.1), (Df = 16), (p = 0.14). There are significant differences between the Colombian and Spanish groups in relation to the EQi-C; chi-Square 7, (Df =16), (p = 0.973). Globally, there is partial scalar invariance for the culture scales, chi-Square 9.158, (Df=8), (p= 0.329); prosocial behavior, chi-Square, 11.246, (Df=7), (p= 0.128); WLEIS, chi-Square 13.444, (Df=7), (p= 0.062); and EQi-C, chi-Square 11.028, (Df=11), (p= 0.441).
Reviewer 3 Report
First of all, thanks for the possibility to review the manuscript,
It's a good work, the authors have listened to the suggestions of the reviewers
With that, the article has improved its writing and understanding
In summary, the manuscript can be published in its current form,
I encourage the authors to continue working in this line.
My congratulations
Author Response
Reviewer 3
Point 1: First of all, thanks for the possibility to review the manuscript. t's a good work, the authors have listened to the suggestions of the reviewers. With that, the article has improved its writing and understanding. In summary, the manuscript can be published in its current form, I encourage the authors to continue working in this line. My congratulations.
Response 1: Thank you very much for your feedback and congratulations.